# Top-Personalized-K Recommendation

Anonymous
anonymous@anonymous

## ABSTRACT

The conventional top-$K$ recommendation, which presents the top-$K$ items with the highest ranking scores, is a common practice for generating personalized ranking lists. However, is this fixed-size top-$K$ recommendation the optimal approach for every user's satisfaction? Not necessarily. We point out that providing fixed-size recommendations without taking into account user utility can be suboptimal, as it may unavoidably include irrelevant items or limit the exposure to relevant ones. To address this issue, we introduce Top-Personalized-$K$ Recommendation, a new recommendation task aimed at generating a personalized-sized ranking list to maximize individual user satisfaction. As a solution to the proposed task, we develop a model-agnostic framework named PerK. PerK estimates the expected user utility by leveraging calibrated interaction probabilities, subsequently selecting the recommendation size that maximizes this expected utility. Through extensive experiments on real-world datasets, we demonstrate the superiority of PerK in Top-Personalized-$K$ recommendation task. We expect that Top-Personalized-$K$ recommendation has the potential to offer enhanced solutions for various real-world recommendation scenarios, based on its great compatibility with existing models.

## CCS CONCEPTS

• **Information systems** → **Collaborative filtering**; **Personalization**; **Recommender systems**.

## KEYWORDS

Recommender System; Collaborative Filtering; Personalization; Recommendation Size; User Utility

**ACM Reference Format:**

Anonymous Author(s). 2024. Top-Personalized-K Recommendation. In *Proceedings of the ACM Web Conference 2024 (WWW '24), May 13-17, 2024, Singapore*. ACM, New York, NY, USA, 11 pages. https://doi.org/XXXXXXX.XXXXXXX

## 1 INTRODUCTION

Personalized recommendations have a significant impact on various daily activities such as shopping, advertising, watching videos, and listening to music. To generate personalized ranking lists of items, recommender systems utilize the top-$K$ recommendation approach [9], which presents the $K$ items with the highest ranking scores,

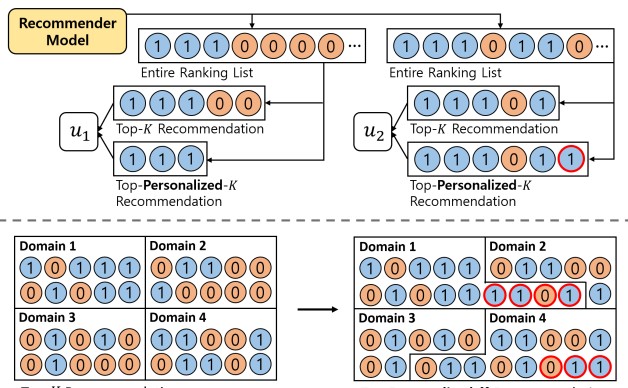

**Figure 1: An example of top-personalized-$K$ recommendation. 1 and 0 represent the relevant and the irrelevant items, respectively. Note that these labels are not available at the inference phase. Newly added items have red outlines.**

sorted in descending order. This approach has become a common practice in recent recommender systems [6, 59, 60] due to its optimality with the globally fixed recommendation size [45]. However, while tremendous efforts have been made on recommender models, an important question has been overlooked in the previous literature: *is the fixed-size top-K recommendation the optimal approach for ensuring every user's satisfaction?*

To elucidate the potential drawbacks of the top-$K$ recommendation approach, we start with a motivating example with $K = 5$ in Figure 1. For user 1, the top-$K$ recommendation *unavoidably includes* two irrelevant items in the tail, as the ranking list could not be filled with enough relevant items. Moreover, for user 2, the recommendation could be further improved by including one additional relevant item in the tail, although user 2 receives a more accurate top-$K$ recommendation with four relevant items. As such, globally fixing the recommendation sizes results in (1) exposing users to irrelevant items that could lead to ad blindness [5] or user attrition [33], and (2) limiting chances to provide relevant items, which can curtail user engagement and revenue [18]. In this context, we argue the top-$K$ recommendation with a globally fixed recommendation size is not the optimal approach for both user satisfaction and system efficacy.

Instead of globally fixed recommendation sizes, a recommendation size that is *personalized* for individual user satisfaction can create enhanced solutions for various recommendation scenarios. Back in Figure 1, by adopting personalized recommendation sizes, the system can increase both users' satisfaction by reducing the effort spent inspecting irrelevant items (user 1) and providing more relevant items (user 2). Furthermore, the personalized recommendation size paves a way to further increase user satisfaction in various applications, especially for systems with limited resources for making recommendations: (1) *Multi-domain recommender systems* [50]

that display items from various domains on a single constrained screen can strike a balance in recommendation sizes for the maximum overall user satisfaction, by employing adapted-sized ranking lists from each domain (Figure 1 bottom). (2) In the case of *sponsored advertisements* [5], advertisers can achieve higher user engagement with the same promotion expenses by adjusting the number of promoted items based on each user's expected utility. (3) In the context of the *prefetching mechanism* [57], which caches the initial few seconds of videos expected to be clicked in order to reduce startup delay, the system can minimize cache size and prevent cache pollution by adjusting the number of items to be cached. In this sense, we claim that embracing personalized recommendation sizes not only enhances user satisfaction but also unlocks diverse optimization possibilities in real-world systems.

In this paper, we propose **Top-Personalized-$K$ Recommendation**, a new recommendation task resolving the limitation of the top-$K$ recommendation. Formally, the top-personalized-$K$ recommendation refers to providing a ranking list of the variable size that maximizes individual user satisfaction, which can be quantitatively measured by user utility [46, 49]. As a solution to the proposed task, we develop **PerK**, a framework to determine the personalized recommendation size with any existing recommender model. PerK first formulates a bi-level optimization problem where the objective is to determine the recommendation size that maximizes each user's utility (Sec 4). To solve this optimization, we introduce the concept of *expected user utility* (Sec 5.2), as it is not feasible to compute the true user utility during the inference phase. We treat the interaction labels of unobserved items as Bernoulli random variables and derive the expectation of the user utility for various widely-used utility measures. Derived expected user utilities can be computed with the interaction probability of user-item pairs.

The remaining challenge is obtaining accurate interaction probability with an arbitrary recommender model. The recommender models do not necessarily output the accurate interaction probability [12, 24]. They often output unbounded ranking scores that cannot be treated as probabilities [13, 44] or miscalibrated interaction probabilities that do not accurately reflect the true likelihood of user-item interactions [14, 28]. To address this problem, we propose the use of *calibrated interaction probability* obtained through the user-wise calibration function (Sec 5.3). The calibration function maps the ranking scores of the recommender model to well-calibrated interaction probabilities [24]. We adopt Platt scaling [40] and instantiate it for each user to consider the different distributions of the ranking score across users. We train the calibration function to predict the interactions between pairs in a held-out calibration set. As a result, the output of the function accurately indicates the true likelihood of interaction, leading to accurate expected user utility.

In summary, we aim to find the optimal recommendation size for each user by (1) obtaining calibrated interaction probability with user-wise calibration, (2) estimating the expected user utility, and (3) determining the recommendation size that results in the maximum expected user utility. The main contributions of our work can be described as follows:

- We highlight the necessity of personalized recommendation size based on its practical advantages in real-world scenarios, which has not been studied well in the previous literature.
- We propose Top-Personalized-$K$ Recommendation, a new recommendation task where the recommendation size can be adjusted for each user to enhance individual user satisfaction.
- We develop PerK, a framework to determine the personalized recommendation size by estimating the user's expected utility with the calibrated interaction probability.
- We conduct comprehensive experiments with three base recommenders on four real-world datasets, demonstrating the superiority of PerK in the top-personalized-$K$ recommendation task.

## 2 RELATED WORK

To the best of our knowledge, personalized recommendation size has not been studied well in the previous literature. The nearest research line is the document list truncation [2, 3] that aims to determine the optimal cutoff position for retrieved documents. Document list truncation has been applied in various domains, including legal document retrieval [31, 51] and searching [53, 58]. The recent methods [3, 29, 53, 58] formulate the truncation problem as a classification task that predicts the optimal position among the candidate cut-off positions. The target label for this classification is given as a $K$-dimensional vector and each element indicates the probability of being the optimal position. Then, they deploy large and deep models (e.g., Bi-LSTM [15], Transformer [52]) for the classification. **Limitations.** While this task is related to ours, directly applying truncation methods to the top-personalized-$K$ recommendation leads to poor performance for several reasons. First, document retrieval datasets [36, 42] have sufficient relevant documents, which provide rich target labels for classification. In contrast, recommendation datasets suffer from severe sparsity and only a small fraction of items is relevant for each user. Additionally, recommendation datasets often include hidden-relevant items among unobserved ones, providing inaccurate and noisy target labels. Second, the document retrieval task leverages high-dimensional features extracted from text contents (e.g., tf-idf [43] or doc2vec [25]) to train large transformer models. However, in our scenarios, these additional features (e.g., review texts) are not always available for unobserved items [7], and scalar ranking scores are insufficient for training large transformer models. As a result, state-of-the-art truncation models (e.g., AttnCut [58] and MtCut [53]) show limited performance when applied to the top-personalized-$K$ task, and a solution tailored to recommender systems is required.

## 3 PRELIMINARIES

### 3.1 Recommendation with Implicit Feedback

**Implicit Feedback.** In this paper, we focus on the recommendation with implicit feedback [16], a widely adopted scenario for top-$K$ recommendation [9]. Let $\mathcal{U}$ and $\mathcal{I}$ denote a set of users and a set of items, respectively. For a pair of $u \in \mathcal{U}$ and $i \in \mathcal{I}$, an interaction label $Y_{u,i}$ is given as 1 if their interaction is observed and 0 otherwise. It is noted that when $Y_{u,i} = 0$, it indicates either that the item is irrelevant to the user or that it can be a *hidden-relevant* item of the user [47]. A dataset $\mathcal{D} = \{(u,i)|Y_{u,i} = 1\}$ consists of positive

pairs, and is split into a training set $\mathcal{D}_{tr}$ and a validation set $\mathcal{D}_{val}$. $\mathcal{I}_u^- = \{i \mid (u, i) \notin \mathcal{D}_{tr}\}$ denotes the unobserved itemset of $u$.

**Recommender Model.** A recommender model $f_\theta : \mathcal{U} \times \mathcal{I} \to \mathbb{R}$ learns to assign a ranking score to each user-item pair. In the literature, a variety of model architectures for $f_\theta$ have been deployed, including matrix factorization [22], neural networks [14, 28], and graph neural networks [13, 59]. To train recommender models, point-wise loss (e.g., binary cross-entropy, mean squared error), pair-wise loss (e.g., BPR [44], Margin Ranking loss [56]), and list-wise loss (e.g., InfoNCE [38], Sampled Softmax [62]) have been adopted. After the recommender model is trained, we produce a ranking list $\pi$ with unobserved items in $\mathcal{I}_u^-$ by sorting the ranking scores $f_\theta(u, i)$ in descending order.

## 3.2 User Utility

In this paper, we adopt user utility for the quantitative measurement of user satisfaction yielded by recommendation [46, 49]. User utility evaluates the ranking list $\pi$ based on how much the recommended items are *exposed* and *relevant* to the user:

$$U(\pi|u) = \sum_{i \in \pi} \omega(\text{rank}(i|\pi))\rho(Y_{u,i}). \tag{1}$$

The function $\omega(\cdot)$ maps an item's rank to the item's exposure based on the position-bias model [8], and $\rho(\cdot)$ casts the relevance of an item to the user utility.[1] $\text{rank}(i|\pi)$ is the rank of item $i$ in the ranking list $\pi$. Depending on the formulation of $\omega(\cdot)$ and $\rho(\cdot)$, we can represent various utility measures. For example, Discounted Cumulative Gain (DCG) [19] can be represented with $\omega(r) = \frac{1}{\log_2(1+r)}$ and $\rho(Y) = \mathbb{1}_{\{Y=1\}}(Y)$. Additionally, other utility measures, such as Normalized Discounted Cumulative Gain [19], Penalized Discounted Cumulative Gain [19, 58], F1 score [41], and Truncated Precision [28, 41] have been adopted in the literature and will be investigated in this work.

## 3.3 Top-K Recommendation

**Definition 1** (Top-K Recommendation). *The top-K recommendation refers to providing a ranking list of K items with the highest ranking scores.*

Typically, the recommendation size ($K$) is globally fixed and predefined by systems, taking into account platform constraints such as screen size, thumbnail dimensions, and promotion expense. The probability ranking principle [45] guarantees that this approach maximizes user utility under a fixed value of $K$ and for any decreasing function of $\omega(\cdot)$ in Eq.1. That is, we get:

$$\pi_K = \underset{i \in \mathcal{I}_u^-}{\text{argsort}} \, f_\theta(u, i)[: K] = \underset{\pi \in \Pi_K}{\text{argmax}} \, U(\pi|u), \tag{2}$$

where $[: K]$ denotes to take the first $K$ elements of the list and $\Pi_K$ is a set of all possible ranking lists, each with a size of $K$. In the rest of this paper, $\pi_K$ represents the sorted top-$K$ ranking list obtained by Eq.2 (e.g., $\pi_m$ denotes the top-$m$ ranking list).

**Limitations.** Despite the prevalence and advancements in the top-$K$ recommendation, as discussed in Section 1, the top-$K$ recommendation has limitations in that it provides a fixed-size recommendation without consideration of user utility. In this case, users must inspect irrelevant items to filter them out, which can be time-consuming, especially in domains with lengthy inspection times (e.g., movies). This process can negatively impact user satisfaction [1], resulting in users ignoring future recommendations [5] or even leaving the system [33]. Moreover, the fixed size scheme may further degrade the efficiency of real-world applications, such as presenting an equal number of items from each domain without taking into account user preferences or promoting an equal number of items to each user without considering the users' expected utility. Nevertheless, the methodology for determining the appropriate number of items to present remains unexplored.

## 4 PROPOSED TASK

We here firstly propose a new recommendation task, named **Top-Personalized-K Recommendation**, as a means to overcome the limitation of the top-$K$ recommendation.

**Definition 2** (Top-Personalized-K Recommendation). *The top-personalized-K recommendation refers to providing a ranking list where the recommendation size is optimized in $[K]$ for each user, to maximize individual user utility.*[2]

This approach ensures that each user receives a tailored-sized recommendation, helps avoid presenting irrelevant items and providing more number of relevant items. The problem of finding the optimal recommendation size $k_{max}$ can be formulated as the following bi-level optimization problem:

$$
\begin{aligned}
k_{max} &= \underset{k \in [K]}{\text{argmax}} \, U(\pi_k|u) \\
\text{s.t.} \quad \pi_k &= \underset{\pi \in \Pi_k}{\text{argmax}} \, U(\pi|u) \\
&= \underset{i \in \mathcal{I}_u^-}{\text{argsort}} \, f_\theta(u, i)[: k], \quad \forall k \in [K].
\end{aligned} \tag{3}
$$

Fortunately, the inner optimization can be done readily since the top-$k$ ranking list $\pi_k$ is the optimal ranking list for each $k$. Then, in the outer optimization, we would like to select the $k$ where $\pi_k$ yields the highest user utility. However, it is noted that *directly computing the user utility is infeasible*, since we do not have access to the true relevance of unobserved items in the inference phase.

**Applications.** While our work primarily focuses on the technical aspects, the proposed task has several implications for real-world applications. Various scenarios can adopt personalized recommendation sizes by modifying the constraint in Eq.3. For instance, in *multi-domain recommender systems* [32, 50], the total number of recommended items from various domains is constrained due to the single limited screen. Instead of displaying the equivalent number of items from each domain, we can adjust the recommendation size of each domain to maximize the overall utility under the constraint (Refer to Appendix A.2 for the modified optimization problem). Similarly, in *sponsored advertisement* [5, 61], the advertiser has a budget constraint on the promotion expenses. In this case, instead of promoting the same number of items to all users, the system can

---

[1]In the evaluation phase, we have the true relevance (i.e., irrelevant or hidden-relevant) for unobserved items.

[2]$[K] = \{1, 2, 3, ..., K\}$

present personalized numbers of promotions depending on each user's utility while still satisfying the global budget constraint.

# 5 PROPOSED FRAMEWORK

## 5.1 Overview

We propose **PerK**, a novel framework to find the optimal recommendation size for the top-personalized-$K$ recommendation. PerK is a model-agnostic framework, allowing it to be adapted for any item recommendation scenario with existing recommenders. To solve the bi-level optimization problem in Eq.3, PerK utilizes:

- **(Sec 5.2) Expected User Utility**: Expected user utility can be estimated by treating the interaction labels for unobserved items as Bernoulli random variables. PerK derives the computational form of the expected user utility for widely-used utility functions, which can be computed with the interaction probabilities.
- **(Sec 5.3) Calibrated Interaction Probability**: To obtain accurate interaction probabilities, PerK utilizes user-wise calibration functions instantiated and trained for each user. The calibration function maps the ranking scores of the recommender model to the calibrated interaction probabilities.

To sum up, given a pre-trained recommender model, (1) PerK trains the user-wise calibration functions and gets the calibrated interaction probabilities for unobserved items (Sec 5.3), (2) PerK estimates the expected user utility for each candidate size $k$ with the calibrated probability (Sec 5.2), (3) PerK selects the size with the maximum expected user utility, and provides the recommendation list having the selected size (Sec 5.4).

## 5.2 Expected User Utility

We cannot compute the true user utility in Eq.3, since we do not have access to the true relevance of unobserved items in the inference phase. To overcome this issue, PerK estimates the **expected user utility** instead of the true value by treating the interaction label $Y$ for unobserved items as Bernoulli random variables. We defined the expected user utility as follows:

$$\mathbb{E}_Y[U(\pi|u)] = \mathbb{E}_Y\Big[\sum_{i\in\pi}\omega(\text{rank}(i|\pi))\rho(Y_{u,i})\Big]. \quad (4)$$

For simplicity, we transform the above formalization for the top-$k$ ranking list $\pi_k$ as follows:

$$\mathbb{E}_Y[U(\pi_k|u)] = \mathbb{E}_Y\Big[\sum_{r=1}^{k}\omega(r)\rho(Y_{u,r})\Big]. \quad (5)$$

With slight abuse of terminology, let $Y_{u,r}$ denote the interaction variable for user $u$ and the $r^{\text{th}}$ item in $\pi_k$. In the rest of this section, we derive the computational form of expected user utility for four widely-adopted utility measures. Due to the lack of space, we cannot provide the complete step-by-step process. **Please refer to Appendix A.1 for a detailed derivation procedure.**

### 5.2.1 *Normalized Discounted Cumulative Gain (NDCG)*. NDCG [19], one of the most established utility measures, is formulated as:

$$U_{\text{NDCG}}(\pi_k|u) = \frac{U_{\text{DCG}}(\pi_k|u)}{U_{\text{IDCG}}(\pi_k|u)} = \frac{\sum_{r=1}^{k}\frac{\mathbb{1}_{\{Y=1\}}(Y_{u,r})}{\log_2(1+r)}}{\sum_{r=1}^{\min(S_Y^u,k)}\frac{1}{\log_2(1+r)}}. \quad (6)$$

$S_Y^u = \sum_{i\in\mathcal{I}_u^-} Y_{u,i}$ is the sum of all interaction variables for unobserved items of user $u$. The expected NDCG with respect to the random variable $Y$ is:

$$\mathbb{E}_Y[U_{\text{NDCG}}(\pi_k|u)] \quad (7a)$$

$$= \sum_{m=1}^{|\mathcal{I}_u^-|} P(S_Y^u = m) \cdot \mathbb{E}_{Y|S_Y^u=m}\left[\frac{\sum_{r=1}^{k}\frac{\mathbb{1}_{\{Y=1\}}(Y_{u,r})}{\log_2(1+r)}}{\sum_{r=1}^{\min(S_Y^u,k)}\frac{1}{\log_2(1+r)}}\right] \quad (7b)$$

$$= \sum_{m=1}^{|\mathcal{I}_u^-|}\frac{\sum_{r=1}^{k}\frac{P(Y_{u,r}=1)P(S_{Y_{-r}}^u=m-1)}{\log_2(1+r)}}{\sum_{r=1}^{\min(m,k)}\frac{1}{\log_2(1+r)}} \quad (7c)$$

$$\approx \sum_{m=1}^{M}\frac{\sum_{r=1}^{k}\frac{P(Y_{u,r}=1)P(S_Y^u=m-1)}{\log_2(1+r)}}{\sum_{r=1}^{\min(m,k)}\frac{1}{\log_2(1+r)}}. \quad (7d)$$

(Eq.7b): Since investigating all possible combinations of $Y_{u,r}$ is intractable, we aggregate the summation over possible $S_Y^u$ by adopting the total expectation theorem [4]. (Eq.7c): $U_{\text{DCG}}(\pi_k|u)$ and $U_{\text{IDCG}}(\pi_k|u)$ are conditionally independent given $S_Y^u$ and $S_{Y_{-r}}^u = S_Y^u - Y_{u,r}$. (Eq.7d): For scalability, we adopt **two simple approximations**: (1) We aggregate the summation only to $M \leq 2000$ rather than $|\mathcal{I}_u^-|$ (here, $M$ is a hyperparameter), since the users are likely to interact with only a few items among the unobserved items. (2) We replace $P(S_{Y_{-r}}^u = m - 1)$ with $P(S_Y^u = m - 1)$ as we confirmed that the effect of one interaction for $S_Y^u$ is negligible. These simple techniques make the expected user utility can be estimated in real-time. $P(S_Y^u = m - 1) = P(\sum_{i\in\mathcal{I}_u^-} Y_{u,i} = m - 1)$ follows the Poisson-Binomial distribution [26], and can be computed only with the interaction probabilities $P(Y_{u,i} = 1)$ for $i \in \mathcal{I}_u^-$.

### 5.2.2 *Penalized Discounted Cumulative Gain (PDCG)*. PDCG [19], a utility measure based on DCG, has a penalizing term for the irrelevant items in the ranking list.[3]

$$U_{\text{PDCG}}(\pi_k|u) = \sum_{r=1}^{k}\frac{\mathbb{1}_{\{Y=1\}}(Y_{u,r}) - \mathbb{1}_{\{Y=0\}}(Y_{u,r})}{\log_2(1+r)}. \quad (8)$$

The expected PDCG with respect to interaction variable $Y$ is computed as follows:

$$\mathbb{E}_Y[U_{\text{PDCG}}(\pi_k|u)] = \mathbb{E}_Y\Big[\sum_{r=1}^{k}\frac{\mathbb{1}_{\{Y=1\}}(Y_{u,r}) - \mathbb{1}_{\{Y=0\}}(Y_{u,r})}{\log_2(1+r)}\Big]$$
$$= \sum_{r=1}^{k}\frac{2\cdot P(Y_{u,r}=1) - 1}{\log_2(1+r)}. \quad (9)$$

### 5.2.3 *F1 Score (F1)*. F1 [41] is a utility measure computed as the harmonic mean of Precision and Recall.

$$U_{\text{F1}}(\pi_k|u) = \frac{2\cdot\frac{\sum_{r=1}^{k}\mathbb{1}_{\{Y=1\}}(Y_{u,r})}{k}\cdot\frac{\sum_{r=1}^{k}\mathbb{1}_{\{Y=1\}}(Y_{u,r})}{S_Y^u}}{\frac{\sum_{r=1}^{k}\mathbb{1}_{\{Y=1\}}(Y_{u,r})}{k} + \frac{\sum_{r=1}^{k}\mathbb{1}_{\{Y=1\}}(Y_{u,r})}{S_Y^u}}$$
$$= \frac{2\cdot\sum_{r=1}^{k}\mathbb{1}_{\{Y=1\}}(Y_{u,r})}{S_Y^u + k}. \quad (10)$$

---

[3]It is called DCG in the document retrieval field [53, 58], however, we call it PDCG to distinguish it from DCG in the item recommendation field in Eq.6.

The expected F1 with respect to the interaction variable $Y$ is computed as follows:

$$
\begin{aligned}
\mathbb{E}_Y[U_{\mathrm{F1}}(\pi_k|u)] &= \sum_{m=1}^{|I_u^-|} P(S_Y^u = m) \frac{2 \cdot \sum_{r=1}^k P(Y_{u,r} = 1|S_Y^u = m)}{m + k} \\
&\approx \sum_{m=1}^{M} \frac{2 \cdot \sum_{r=1}^k P(Y_{u,r} = 1)P(S_Y^u = m - 1)}{m + k}.
\end{aligned}
\tag{11}
$$

Here, we use the total expectation theorem and apply the same approximations as done in Eq.7.

*5.2.4  **Truncated Precision (TP)**.* TP [28, 41] is a utility measure that addresses the limitations of Recall and Precision.[4]

$$
U_{\mathrm{TP}}(\pi_k|u) = \frac{\sum_{r=1}^k \mathbb{1}_{\{Y=1\}}(Y_{u,r})}{\min(k, S_Y^u)}.
\tag{12}
$$

The expected TP with respect to the interaction variable $Y$ is computed as follows:

$$
\begin{aligned}
\mathbb{E}_Y[U_{\mathrm{TP}}(\pi_k|u)] &= \sum_{m=1}^{|I_u^-|} P(S_Y^u = m) \frac{\sum_{r=1}^k P(Y_{u,r} = 1|S_Y^u = m)}{\min(k, m)} \\
&\approx \sum_{m=1}^{M} \frac{\sum_{r=1}^k P(Y_{u,r} = 1)P(S_Y^u = m - 1)}{\min(k, m)}.
\end{aligned}
\tag{13}
$$

Here, we use the total expectation theorem and apply the same approximations as done in Eq.7.

*5.2.5  **Interaction Probability**.* Up to this point, we have formulated the computational forms of expected user utility for four widely-adopted utility measures: NDCG (Eq.7), PDCG (Eq.9), F1 (Eq.11), and TP (Eq.13). In common, these measures all require the interaction probability $P(Y_{u,r} = 1)$ for the estimation. In the following subsection, we present our solution to obtain accurate interaction probabilities with an arbitrary recommender model.

## 5.3  Calibrated Interaction Probability

Recommender models do not necessarily output accurate interaction probability. They often output the ranking score that can have any value of an unbounded real number [13, 44, 59], making it difficult to treat it as a probability. Furthermore, even when a model is trained to output probabilities [14, 28], it has been demonstrated that these probabilities may not accurately reflect the true likelihood (i.e., model miscalibration) [12, 24].

To address this, we introduce a post-processing calibration function $g_\phi : \mathbb{R} \to [0, 1]$ that maps the ranking score $s_{u,i} = f_\theta(u, i)$ of a pre-trained recommender to the calibrated interaction probability $g_\phi(s_{u,i}) = \hat{P}(Y_{u,i} = 1|s_{u,i})$. A probability $p$ is regarded *calibrated* if it indicates the ground-truth correctness likelihood [23]: $\mathbb{E}[Y|g_\phi(s) = p] = p$. For example, if we have 100 user-item pairs with $p = 0.2$, we expect 20 of them to have interactions ($Y = 1$).

We adopt Platt scaling $g_\phi(s) = \sigma(as + b)$ [40], a generalized form of temperature scaling [12]. This calibration function has been deployed effectively for model calibration in computer vision

---

**Algorithm 1:** Top-Personalized-K Recommendation with PerK

**Input**  : Training set $\mathcal{D}_{\mathrm{tr}}$, validation set $\mathcal{D}_{\mathrm{val}}$, pre-trained recommender model $f_\theta$, maximum recommendation size $K$, expected user utility $\mathbb{E}_Y[U(\pi_k|u)]$

**Output**: Top-personalized-$K$ recommendation $\pi^{\mathrm{perK}}$

1 Fit user-wise calibration $g_{\phi_u}$ on $f_\theta$ ▷ Section 5.3

2 $\pi_K = \mathrm{argsort}_{i \in I_u^-} f_\theta(u, i)[: K]$ ▷ Top-$K$ Recommendation

3 **for** $k \in [K]$ **do**

4     $\pi_k = \pi_K[: k]$

5     Compute $\mathbb{E}_Y[U(\pi_k|u)]$ with $g_{\phi_u}$ ▷ Section 5.2

6 $k_{\max} = \mathrm{argmax}_{k \in [K]} \mathbb{E}_Y[U(\pi_k|u)]$.

7 $\pi^{\mathrm{perK}} = \pi_{k_{\max}}$

---

[11, 34], natural language processing [10], and recommender system [24]. The key difference is that PerK instantiates the calibration function for each user, while previous calibration work [24] deploys one global calibration function covering all users. That is, we have:

$$
g_{\phi_u}(s) = \sigma(a_u s + b_u), \quad \forall u \in \mathcal{U}.
\tag{14}
$$

The user-specific parameters $\phi_u = \{a_u, b_u\}$ are related to the distribution of the ranking score of each user [23, 24]. Therefore, the user-wise calibration function can consider the different distributions of the ranking score across users.

We train the calibration function to predict the interactions of pairs in a calibration set constructed for each user. It is a common practice to adopt the validation set as the calibration set [12, 24]. In this work, the calibration set for user $u$ is constructed as follows:

$$
\mathcal{D}_u^{\mathrm{cal}} = \{(u, i, Y_{u,i} = 1)|i \in I_u^{\mathrm{val}}\} \cup \{(u, i, Y_{u,i} = 0)|i \in I_u^- \setminus I_u^{\mathrm{val}}\},
\tag{15}
$$

where $I_u^{\mathrm{val}} = \{i|(u, i) \in \mathcal{D}_{\mathrm{val}}\}$. We also use the binary cross-entropy loss, a widely-adopted loss function for the calibration of binary classifiers [12, 23, 24], as our calibration loss.

$$
\mathcal{L}_u^{\mathrm{cal}} = \sum_{(u,i,Y_{u,i}) \in \mathcal{D}_u^{\mathrm{cal}}} -Y_{u,i}\log(g_{\phi_u}(s_{u,i})) - (1 - Y_{u,i})\log(1 - g_{\phi_u}(s_{u,i})).
\tag{16}
$$

During the fitting of the calibration function, the base recommender model $f_\theta$ is fixed and only $g_{\phi_u}$ is updated. It is worth mentioning that Platt scaling with binary cross-entropy is mathematically equivalent to logistic regression and can be efficiently solved. Additionally, as we only have two learnable parameters for each user, our calibration functions have a negligible impact on space complexity.

## 5.4  Optimization Procedure

Algorithm 1 presents the entire procedure of PerK solving the bi-level optimization in Eq.3 for the top-personalized-$K$ recommendation. (line 1): PerK first fits the user-wise calibration function $g_{\phi_u}$ with $\mathcal{D}_u^{\mathrm{cal}}$ and $\mathcal{L}_u^{\mathrm{cal}}$ on top of pre-trained and fixed recommender $f_\theta$. (line 2): PerK generates top-$K$ recommendation by sorting the ranking score. (line 3-5): For each candidate size $k$, PerK estimates the

---

[4]For example, Precision cannot have a value of 1 if $k$ is larger than $S_Y^u$. TP is referred to as Recall in [28]. However, in this work, we use the term Truncated Precision to distinguish it from the standard definition of Recall.

**Table 1: Average user utility of recommendations produced by compared methods when they are optimized for each target utility measure. *Improv.* denotes the improvement of PerK over MtCut. The numbers in boldface denote the best result in each setting. *We conduct the paired t-test with a 0.05 level and every *Improv.* is statistically significant.**

| Base Model | Method | MovieLens 10M | | | | CiteULike | | | | MovieLens 25M | | | | Amazon Books | | | |
|---|---|---|---|---|---|---|---|---|---|---|---|---|---|---|---|---|---|
| | | NDCG | PDCG | F1 | TP | NDCG | PDCG | F1 | TP | NDCG | PDCG | F1 | TP | NDCG | PDCG | F1 | TP |
| BPR | Oracle | 0.6190 | 0.6702 | 0.3152 | 0.7078 | 0.3020 | -0.3677 | 0.1806 | 0.3940 | 0.5961 | 0.6089 | 0.2883 | 0.6812 | 0.1895 | -0.7438 | 0.1133 | 0.2643 |
| | Top-1 | 0.4549 | -0.0902 | 0.0546 | 0.4549 | 0.1837 | -0.6326 | 0.0338 | 0.1837 | 0.4392 | -0.1216 | 0.0492 | 0.4392 | 0.0961 | -0.8078 | 0.0216 | 0.0961 |
| | Top-5 | 0.3925 | -0.6428 | 0.1541 | 0.3751 | 0.1489 | -2.0787 | 0.0842 | 0.1401 | 0.3729 | -0.7561 | 0.1357 | 0.3547 | 0.0786 | -2.4938 | 0.0507 | 0.0741 |
| | Top-10 | 0.3725 | -1.3715 | 0.1984 | 0.3584 | 0.1495 | -3.3650 | 0.1024 | 0.1483 | 0.3496 | -1.5372 | 0.1757 | 0.3328 | 0.0828 | -3.9401 | 0.0582 | 0.0852 |
| | Top-20 | 0.3734 | -2.8434 | 0.2228 | 0.3849 | 0.1668 | -5.4853 | 0.1108 | 0.1907 | 0.3460 | -3.0494 | 0.2013 | 0.3510 | 0.0955 | -6.2660 | 0.0584 | 0.1155 |
| | Top-50 | 0.4057 | -7.1636 | 0.2117 | 0.4841 | 0.2021 | -10.8747 | 0.0930 | 0.2834 | 0.3740 | -7.3813 | 0.1983 | 0.4421 | 0.1203 | -11.8590 | 0.0496 | 0.1826 |
| | Rand | 0.3872 | -3.6387 | 0.2046 | 0.4144 | 0.1738 | -6.2622 | 0.1003 | 0.2058 | 0.3611 | -3.8266 | 0.1865 | 0.3801 | 0.0998 | -7.1022 | 0.0535 | 0.1284 |
| | Val-$k$ | 0.4562 | -0.0640 | 0.2270 | 0.4906 | 0.1922 | -0.6162 | 0.1061 | 0.2266 | 0.4368 | -0.0532 | 0.2052 | 0.4347 | 0.1093 | -0.8378 | 0.0563 | 0.1362 |
| | AttnCut | 0.4604 | -0.0702 | 0.2371 | 0.4969 | 0.2024 | -0.6142 | 0.1063 | 0.2835 | 0.4392 | -0.1214 | 0.2087 | 0.4392 | 0.1203 | -0.8078 | 0.0566 | 0.1826 |
| | MtCut | 0.4621 | -0.0167 | 0.2383 | 0.5037 | 0.2034 | -0.6022 | 0.1071 | 0.2838 | 0.4413 | -0.0616 | 0.2140 | 0.4639 | 0.1212 | -0.8078 | 0.0568 | 0.1826 |
| | PerK | **0.4901** | **0.2087** | **0.2538** | **0.5711** | **0.2159** | **-0.4971** | **0.1117** | **0.2993** | **0.4687** | **0.1876** | **0.2300** | **0.5401** | **0.1261** | **-0.7952** | **0.0619** | **0.1894** |
| | *Improv.* | 6.06%* | - | 6.50%* | 13.38%* | 6.15%* | - | 4.30%* | 5.46%* | 6.21%* | - | 7.48%* | 16.43%* | 4.04%* | - | 8.98%* | 3.72%* |
| NCF | Oracle | 0.5760 | 0.4792 | 0.2896 | 0.6688 | 0.2813 | -0.4076 | 0.1685 | 0.3690 | 0.5310 | 0.3544 | 0.2513 | 0.6157 | 0.1444 | -0.8246 | 0.0885 | 0.2076 |
| | Top-1 | 0.4081 | -0.1839 | 0.0474 | 0.4081 | 0.1700 | -0.6601 | 0.0304 | 0.1700 | 0.3761 | -0.2479 | 0.0393 | 0.3761 | 0.0666 | -0.8668 | 0.0150 | 0.0666 |
| | Top-5 | 0.3521 | -0.8801 | 0.1350 | 0.3368 | 0.1403 | -2.1273 | 0.0786 | 0.1330 | 0.3206 | -1.0628 | 0.1117 | 0.3055 | 0.0563 | -2.6231 | 0.0370 | 0.0537 |
| | Top-10 | 0.3360 | -1.6822 | 0.1764 | 0.3251 | 0.1384 | -3.4438 | 0.0942 | 0.1368 | 0.3011 | -1.9434 | 0.1474 | 0.2873 | 0.0606 | -4.1048 | 0.0434 | 0.0638 |
| | Top-20 | 0.3397 | -3.2150 | 0.2023 | 0.3545 | 0.1529 | -5.5941 | 0.1015 | 0.1733 | 0.2980 | -3.5640 | 0.1721 | 0.3033 | 0.0712 | -6.4683 | 0.0443 | 0.0888 |
| | Top-50 | 0.3721 | -7.6021 | 0.1966 | 0.4525 | 0.1866 | -11.0026 | 0.0870 | 0.2618 | 0.3237 | -8.0277 | 0.1744 | 0.3870 | 0.0916 | -12.1148 | 0.0384 | 0.1437 |
| | Rand | 0.3527 | -3.9939 | 0.1866 | 0.3821 | 0.1609 | -6.4157 | 0.0910 | 0.1884 | 0.3102 | -4.3422 | 0.1603 | 0.3298 | 0.0749 | -7.3212 | 0.0407 | 0.0987 |
| | Val-$k$ | 0.4123 | -0.1030 | 0.2071 | 0.4554 | 0.1794 | -0.5801 | 0.0989 | 0.2101 | 0.3769 | -0.5071 | 0.1729 | 0.3842 | 0.0791 | -0.8935 | 0.0422 | 0.1018 |
| | AttnCut | 0.4154 | -0.0839 | 0.2068 | 0.4612 | 0.1812 | -0.5527 | 0.0991 | 0.2619 | 0.3762 | -0.1166 | 0.1755 | 0.3994 | 0.0918 | -0.8668 | 0.0437 | 0.1438 |
| | MtCut | 0.4195 | 0.0805 | 0.2153 | 0.4798 | 0.1891 | -0.5499 | 0.1006 | 0.2627 | 0.3798 | -0.0142 | 0.1837 | 0.4079 | 0.0923 | -0.8667 | 0.0441 | 0.1441 |
| | PerK | **0.4482** | **0.0928** | **0.2293** | **0.5335** | **0.2061** | **-0.5177** | **0.1080** | **0.2764** | **0.4056** | **0.0047** | **0.1960** | **0.4742** | **0.0982** | **-0.8646** | **0.0468** | **0.1553** |
| | *Improv.* | 6.84%* | - | 6.50%* | 11.19%* | 8.99%* | - | 7.36%* | 5.22%* | 6.79%* | - | 6.70%* | 16.25%* | 6.39%* | - | 6.12%* | 7.77%* |
| LightGCN | Oracle | 0.6249 | 0.7459 | 0.3094 | 0.7051 | 0.3471 | -0.2352 | 0.2104 | 0.4400 | 0.5893 | 0.4840 | 0.2864 | 0.6772 | 0.1782 | -0.7571 | 0.1061 | 0.2454 |
| | Top-1 | 0.4749 | -0.0502 | 0.0565 | 0.4749 | 0.2182 | -0.5636 | 0.0420 | 0.2182 | 0.4236 | -0.1527 | 0.0490 | 0.4236 | 0.0919 | -0.8162 | 0.0213 | 0.0919 |
| | Top-5 | 0.4004 | -0.5959 | 0.1531 | 0.3799 | 0.1829 | -1.8801 | 0.1068 | 0.1735 | 0.3603 | -0.8313 | 0.1349 | 0.3427 | 0.0744 | -2.5187 | 0.0487 | 0.0699 |
| | Top-10 | 0.3763 | -1.3345 | 0.1952 | 0.3584 | 0.1829 | -3.1084 | 0.1270 | 0.1816 | 0.3391 | -1.6383 | 0.1741 | 0.3237 | 0.0783 | -3.9772 | 0.0552 | 0.0803 |
| | Top-20 | 0.3734 | -2.8186 | 0.2180 | 0.3793 | 0.1999 | -5.1979 | 0.1303 | 0.2244 | 0.3379 | -3.1805 | 0.1988 | 0.3449 | 0.0898 | -6.3212 | 0.0543 | 0.1077 |
| | Top-50 | 0.3997 | -7.1674 | 0.2068 | 0.4673 | 0.2354 | -10.5738 | 0.1037 | 0.3174 | 0.3687 | -7.5533 | 0.1951 | 0.4396 | 0.1117 | -11.9466 | 0.0449 | 0.1663 |
| | Rand | 0.3875 | -3.5835 | 0.2001 | 0.4062 | 0.2075 | -6.0557 | 0.1182 | 0.2450 | 0.3527 | -3.9406 | 0.1839 | 0.3735 | 0.0938 | -7.1826 | 0.0497 | 0.1187 |
| | Val-$k$ | 0.4652 | 0.0417 | 0.2225 | 0.4894 | 0.2296 | -0.4288 | 0.1286 | 0.2620 | 0.4253 | -0.0864 | 0.1953 | 0.4439 | 0.1017 | -0.8469 | 0.0532 | 0.1247 |
| | AttnCut | 0.4749 | -0.0412 | 0.2237 | 0.4899 | 0.2260 | -0.5616 | 0.1291 | 0.3186 | 0.4242 | -0.1527 | 0.2041 | 0.4359 | 0.1121 | -0.8162 | 0.0516 | 0.1707 |
| | MtCut | 0.4761 | 0.1171 | 0.2318 | 0.4949 | 0.2369 | -0.5487 | 0.1319 | 0.3223 | 0.4317 | 0.0649 | 0.2113 | 0.4440 | 0.1185 | -0.8162 | 0.0536 | 0.1761 |
| | PerK | **0.4993** | **0.3309** | **0.2489** | **0.5702** | **0.2551** | **-0.3989** | **0.1383** | **0.3438** | **0.4543** | **0.0876** | **0.2277** | **0.4742** | **0.1261** | **-0.7912** | **0.0571** | **0.1878** |
| | *Improv.* | 4.87%* | - | 7.38%* | 15.22%* | 7.68%* | - | 4.85%* | 6.67%* | 5.24%* | - | 7.76%* | 6.80%* | 6.41%* | - | 6.53%* | 6.64%* |

expected user utility $\mathbb{E}_Y[U(\pi_k|u)]$ by using the calibrated interaction probability $g_{\phi_u}(s_{u,i}) = \hat{P}(Y_{u,i} = 1|s_{u,i})$. (line 6-7): Lastly, PerK select $k_{\max}$, the recommendation size with the highest expected user utility, and provide the top-personalized-$K$ recommendation $\pi^{\text{perK}} = \pi_{k_{\max}}$ to user $u$.

## 6 EXPERIMENTS

### 6.1 Experiment Setup

We provide a summary of the experiment setup due to limited space. Please refer to Appendix A.3 for more details. We will publicly provide the GitHub repository of this work in the final version.

*6.1.1 Datasets.* We use four real-world datasets including Movie-Lens 10M,[5] CiteULike [54], MovieLens 25M, and Amazon Books [37]. These datasets are publicly available and have been widely used in the literature [13, 14, 27, 55]. We adopt the 20-core setting

---
[5]https://grouplens.org/datasets/movielens/

for all datasets as done in MovieLens datasets. We randomly split each user's interactions into a training set (60%), a validation set (20%), and a test set (20%).

*6.1.2 Methods Compared.* We compare PerK with various traditional and recent methods. Specifically, we adopt

- **Oracle**: It uses the ground-truth labels of the test set to determine the optimal recommendation size for each user.

and three traditional methods:

- **Top-$k$**: It denotes the top-$k$ recommendation with globally fixed recommendation size. We adopt $k \in \{1, 5, 10, 20, 50\}$.
- **Rand**: It randomly selects the recommendation size for each user. It represents the lower bound of the performance.
- **Val-$k$**: It selects the recommendation size that maximizes validation utility for each user.

and two state-of-the-art methods for truncating document retrieval results:

- **AttnCut** [58]: It deploys a classification model with Bi-LSTM and Transformer encoder to predict the best cut-off position.
- **MtCut** (MMoECut) [53]: It deploys MMoE [30] on top of AttnCut architecture and adopt multi-task learning.

and the proposed framework:

- **PerK (ours)**: We determine the personalized recommendation size by estimating the expected user utility with calibrated interaction probability.

For a quantitative comparison of user utility, we set the maximum recommendation size to 50 for all methods. We also tried 100 and observed a similar performance improvement for PerK.

*6.1.3 Base Recommender Models ($f_\theta$).* We adopt three widely-used recommender models with various model architectures and loss functions: BPR [44], NCF [14], and LightGCN [13]. The ranking score from the base recommender model serves as input for PerK, AttnCut, and MtCut.

## 6.2 Comparison of User Utility

Table 1 presents the average user utility yielded by the recommendation of each compared method. *For a fair comparison, we adopt the same base model and the same target utility throughout the training of AttnCut, MtCut, and PerK in each experimental configuration.* We report the average result of three independent runs.

**Benefits of Top-Personalized-$K$ Recommendation.** We first observe that personalizing the recommendation size results in higher user utility than the globally fixed size. Oracle shows significantly higher utility compared to the best value of Top-$k$. This upper bound on user utility highlights the importance of the proposed task in improving user satisfaction. Accordingly, methods determining the personalized recommendation sizes (i.e., AttnCut, MtCut, and PerK) generally outperform Top-$k$. Furthermore, we notice that Top-$k$ shows a large performance deviation depending on the recommended size. Therefore, the system should avoid naively setting a globally fixed recommendation size and instead determine the personalized size for improved user satisfaction.

**Effectiveness of PerK.** We observe that PerK outperforms the competitors in the top-personalized-$K$ recommendation task. Val-$k$ does not perform well since overfitting to the validation utility is not effective due to the sparse and noisy interactions. Furthermore, the effectiveness of the methods for document list truncation (i.e., AttnCut and MtCut) remains limited, since the target label for the classification does not provide enough supervision due to the hidden-positive and noisy interactions. Indeed, we discovered that they can be easily overfitted to select the size that works well globally (Figure 2), resulting in comparable performance to the best of Top-$k$. In contrast, PerK does not rely on a deep model and considers the hidden-relevant items, to estimate the expected user utility. PerK directly computes the expected user utility in mathematical form with the calibrated interaction probability and significantly outperforms AttnCut and MtCut in the top-personalized-$K$ task.

## 6.3 Personalized Recommendation Size

Figure 2 shows the distributions of recommendation sizes determined by Oracle, PerK, and MtCut. The base model is BPR and the target user utility is F1. We have the following findings: (1) The

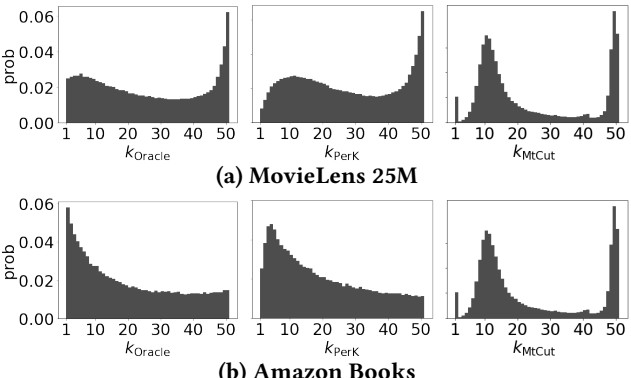

**(a) MovieLens 25M**

**(b) Amazon Books**

**Figure 2: Distribution of recommendation sizes from each method. Three figures in the same row share the y-axis.**

**Table 2: Calibration error and user utility for each calibration method. User utilities are computed when each calibration method is applied to PerK. Lower is better for ECE.**

| Dataset | Calibration | ECE ↓ | NDCG | PDCG | F1 | TP |
|---|---|---|---|---|---|---|
| ML10M | uncalibrated | 0.1284 | 0.4083 | -0.1071 | 0.1966 | 0.4481 |
| | global | 0.0046 | 0.4255 | 0.0838 | 0.2175 | 0.4925 |
| | user-wise | **0.0011** | **0.4482** | **0.0928** | **0.2293** | **0.5335** |
| CiteULike | uncalibrated | 0.0480 | 0.1811 | -0.5667 | 0.0975 | 0.2512 |
| | global | 0.0017 | 0.1940 | -0.5306 | 0.1054 | 0.2639 |
| | user-wise | **0.0003** | **0.2061** | **-0.5177** | **0.1080** | **0.2764** |
| ML25M | uncalibrated | 0.1572 | 0.3612 | -0.2311 | 0.1669 | 0.3812 |
| | global | 0.0422 | 0.3761 | -0.1015 | 0.1757 | 0.4007 |
| | user-wise | **0.0098** | **0.4056** | **0.0047** | **0.1960** | **0.4742** |
| ABooks | uncalibrated | 0.3371 | 0.0812 | -0.8864 | 0.0427 | 0.1391 |
| | global | 0.0581 | 0.0946 | -0.8655 | 0.0449 | 0.1477 |
| | user-wise | **0.0171** | **0.0982** | **-0.8646** | **0.0468** | **0.1553** |

distributions of Oracle show that the optimal recommendation size for maximum user utility differs for each user. (2) The distributions of MtCut are severely skewed towards $k \in [10, 20]$ and have a high peak in that range. The reason is that MtCut is overfitted to select the globally well-performing $k$ which falls into the range of [10, 20] (refer to F1 on MovieLens 25M and Amazon Books with BPR in Table 1). (3) The distributions of PerK are smooth and fairly close to those of Oracle. It is noted that we can set the constraints for the minimum recommendation size in Eq.3, if the system requires it.

## 6.4 Ablation Study for Calibration

Table 2 presents the ablation study on the calibration method with NCF [14] as a base model. We compare (1) the calibration performance, and (2) user utility when it is applied to PerK. The calibration performance is measured by Expected Calibration Error (ECE) [35], a widely used metric for measuring the gap between the output probability and true likelihood of interaction [12, 24]. We observe that the proposed user-wise calibration function shows lower ECE than the global calibration function. Accordingly, PerK yields higher user utilities when it adopts user-wise calibration, demonstrating the superiority of the proposed user-wise calibration over the global calibration.

**Table 3: Space and Time analysis. #Params. denotes the number of learnable parameters and Time denotes the wall time (in ms) used for generating a ranking list for a user.**

| Base model | Method | ML10M | | CiteULike | | ML25M | | ABooks | |
|---|---|---|---|---|---|---|---|---|---|
| | | #Params. | Time | #Params. | Time | #Params. | Time | #Params. | Time |
| BPR | Top-k | 5.041M | 2.871 | 1.291M | 2.086 | 11.574M | 2.374 | 34.542M | 2.636 |
| | PerK | 5.181M | 3.022 | 1.292M | 2.241 | 11.898M | 2.652 | 34.857M | 2.973 |
| NCF | Top-k | 10.092M | 4.092 | 2.581M | 2.566 | 23.156M | 3.067 | 69.117M | 7.767 |
| | PerK | 10.232M | 4.406 | 2.588M | 2.817 | 23.480M | 3.378 | 69.432M | 8.871 |
| LightGCN | Top-k | 5.042M | 3.195 | 2.571M | 2.493 | 23.147M | 2.628 | 34.542M | 3.781 |
| | PerK | 5.181M | 3.449 | 2.577M | 2.759 | 23.472M | 2.969 | 34.857M | 4.106 |

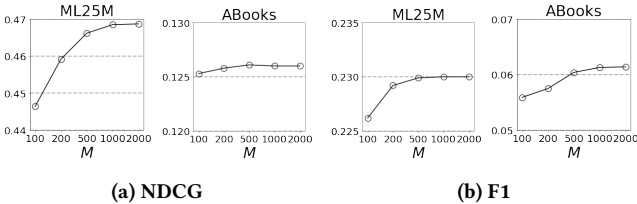

(a) NDCG                                              (b) F1

**Figure 3: Hyperparameter study of $M$**

## 6.5 Space and Time analysis

Table 3 shows the number of learnable parameters and inference time of Top-$k$ and PerK. The target user utility is NDCG for PerK. We use PyTorch [39] with CUDA on GTX Titan Xp GPU and Intel Xeon(R) E5-2640 v4 CPU. First, PerK does not significantly increase the number of learnable parameters from Top-$k$. PerK only has two additional parameters for each user's calibration function, and it has a negligible impact considering the size of the user embedding is typically selected in the range of 64-128. Second, the inference time is increased by about 10%. To speed up the estimation of expected user utility, (1) We perform the user-wise calibration for all users together with a few matrix operations and estimate the expected user utility with various $k$ in a parallel way, and (2) We adopt two approximation techniques for fast computation of the expected user utility in Eq.7d. The hyperparameter study for $M$, which is used for this approximation, with BPR is presented in Figure 3. We can see that it is enough to aggregate the summation in Eq.7d just to $M = 2000$ rather than to the number of all unobserved items $|\mathcal{I}_u^-|$.

## 6.6 Case Study

We present a case study on Amazon dataset to provide concrete examples of how PerK can be applied in real-world applications.

**Single-domain scenario.** Figure 4 (a) shows a case study of the single-domain scenario on Amazon Books with BPR as a base model and F1 as a target utility. Real-world recommender systems often display trending items alongside personalized ones, to create a balanced experience that encompasses both popular choices and individual preferences [17]. We determine the number of personalized items with PerK, and the remaining slots are then populated with trending items. As a result, F1 for the personalized items increases by a large margin for both users, and more trending items can be presented to the second user. In this context, the top-personalized-$K$ recommendation allows the system to strike a balance between the exploration-exploitation trade-off by effectively adjusting the number of personalized items.

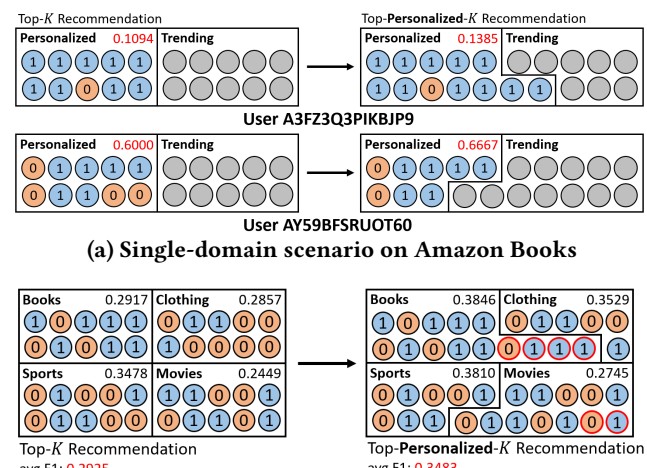

(a) Single-domain scenario on Amazon Books

(b) Multi-domain scenario on Amazon

**Figure 4: Case study. 1 and 0 represent the relevant and the irrelevant labels, which are unavailable at the inference phase. Decimal numbers indicate the F1 score.**

**Multi-domain scenario.** Figure 4 (b) shows a case study of the multi-domain scenario on the four largest domains of Amazon datasets with BPR as a base model and F1 as a target utility. Here, we have a constraint on the total recommendation size considering the system's limited resources, such as screen size and thumbnail dimensions. We generate top-personalized-$K$ recommendations for each domain by slightly modifying Eq.3, to maximize the average F1 across all domains under the constraint that the total recommendation size should not exceed 40 (The modified optimization problem for this scenario is presented in Appendix A.2). As a result, we can see that the average F1 score increases as we adopt the top-personalized-$K$ recommendation scheme. This example demonstrates that the top-personalized-$K$ recommendation can be adopted to multi-domain systems for displaying the optimized number of items from each domain on a single constrained page.

## 7 CONCLUSION

We first highlight the necessity of personalized recommendation size based on its practical advantages in real-world scenarios, which has not been studied well in the previous literature. Then, we propose Top-Personalized-$K$ Recommendation, a new recommendation task that aims to find the optimal recommendation size for each user to maximize individual user satisfaction. As a solution to the top-personalized-$K$ recommendation, we propose PerK, a framework determining the recommendation size that maximizes the expected user utility estimated by using calibrated interaction probabilities. In our thorough experiments on real-world datasets, PerK outperforms recent competitors in the top-personalized-$K$ recommendation task. We believe that the top-personalized-$K$ recommendation can provide enhanced solutions for various item recommendation scenarios and anticipate future work on applications including multi-domain recommender systems, sponsored advertisements, and prefetching mechanisms.

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

# A APPENDIX

## A.1 Derivation of Expected User Utility

### A.1.1 *Normalized Discounted Cumulative Gain (NDCG)*.
The expected NDCG with respect to the random variable $Y$ is:

$$
\begin{aligned}
\mathbb{E}_Y[U_{\text{NDCG}}(\pi_k|u)] &= \mathbb{E}_Y\left[\frac{U_{\text{DCG}}(\pi_k|u)}{U_{\text{IDCG}}(\pi_k|u)}\right] \\
&= \sum_{Y_{u,1},\ldots,Y_{u,|\mathcal{I}_u^-|}} P(Y_{u,1},\ldots,Y_{u,|\mathcal{I}_u^-|}) \frac{U_{\text{DCG}}(\pi_k|u)|_{Y_{u,*}}}{U_{\text{IDCG}}(\pi_k|u)|_{Y_{u,*}}}.
\end{aligned}
\tag{17}
$$

$U_{\text{DCG}}(\pi_k|u)$ and $U_{\text{IDCG}}(\pi_k|u)$ are conditionally independent given $Y_{u,*}$. Since investigating all possible combinations of $Y_{u,r}$ is intractable, we re-formulate the above equation with the summation over possible $S_Y^u$ by adopting the total expectation theorem [4].

$$
\begin{aligned}
\mathbb{E}_Y[U_{\text{NDCG}}(\pi_k|u)] &= \sum_{m=1}^{|\mathcal{I}_u^-|} P(S_Y^u = m)\, \mathbb{E}_{Y|S_Y^u=m}\left[\frac{U_{\text{DCG}}(\pi_k|u)}{U_{\text{IDCG}}(\pi_k|u)}\right] \\
&= \sum_{m=1}^{|\mathcal{I}_u^-|} P(S_Y^u = m) \frac{\mathbb{E}_{Y|S_Y^u=m}[U_{\text{DCG}}(\pi_k|u)]}{U_{\text{IDCG}}(\pi_k|u)|_{S_Y^u=m}}.
\end{aligned}
\tag{18}
$$

$U_{\text{DCG}}(\pi_k|u)$ and $U_{\text{IDCG}}(\pi_k|u)$ are conditionally independent given $S_Y^u$, and $U_{\text{IDCG}}(\pi_k|u)|_{S_Y^u=m} = \sum_{r=1}^{\min(m,k)} \frac{1}{\log(1+r)}$. The expected DCG with respect to $Y$ conditioned on $S_Y^u = m$ can be computed as follows:

$$
\begin{aligned}
\mathbb{E}_{Y|S_Y^u=m}[U_{\text{DCG}}(\pi_k|u)] &= \mathbb{E}_{Y|S_Y^u=m}\left[\sum_{r=1}^{k} \frac{\mathbb{1}_{\{Y=1\}}(Y_{u,r})}{\log_2(1+r)}\right] \\
&= \sum_{r=1}^{k} \frac{P(Y_{u,r}=1|S_Y^u=m)}{\log_2(1+r)} \\
&= \sum_{r=1}^{k} \frac{P(Y_{u,r}=1)P(S_{Y_{-r}}^u=m-1)}{P(S_Y^u=m)\log_2(1+r)},
\end{aligned}
\tag{19}
$$

where $S_{Y_{-r}}^u = S_Y^u - Y_{u,r}$. After aggregating Eq.18 and Eq.19, we get

$$
\mathbb{E}_Y[U_{\text{NDCG}}(\pi_k|u)] = \sum_{m=1}^{|\mathcal{I}_u^-|} \frac{\sum_{r=1}^{k} \frac{P(Y_{u,r}=1)P(S_{Y_{-r}}^u=m-1)}{\log_2(1+r)}}{\sum_{r=1}^{\min(m,k)} \frac{1}{\log_2(1+r)}}.
\tag{20}
$$

For scalability, we adopt **two simple approximations** and get

$$
\mathbb{E}_Y[U_{\text{NDCG}}(\pi_k|u)] \approx \sum_{m=1}^{M} \frac{\sum_{r=1}^{k} \frac{P(Y_{u,r}=1)P(S_Y^u=m-1)}{\log_2(1+r)}}{\sum_{r=1}^{\min(m,k)} \frac{1}{\log_2(1+r)}}.
\tag{21}
$$

The details about the approximations are presented in Sec 5.2.1.

### A.1.2 *F1 Score (F1)*.
The expected F1 with respect to the interaction variable $Y$ is computed as follows:

$$
\begin{aligned}
\mathbb{E}_Y[U_{\text{F1}}(\pi_k|u)] &= \sum_{m=1}^{|\mathcal{I}_u^-|} P(S_Y^u = m)\, \mathbb{E}_{Y|S_Y^u=m}[U_{\text{F1}}(\pi_k|u)] \\
&= \sum_{m=1}^{|\mathcal{I}_u^-|} P(S_Y^u = m) \frac{2 \cdot \sum_{r=1}^{k} P(Y_{u,r}=1|S_Y^u=m)}{m+k} \\
&= \sum_{m=1}^{|\mathcal{I}_u^-|} \frac{2 \cdot \sum_{r=1}^{k} P(Y_{u,r}=1)P(S_{Y_{-r}}^u=m-1)}{m+k} \\
&\approx \sum_{m=1}^{M} \frac{2 \cdot \sum_{r=1}^{k} P(Y_{u,r}=1)P(S_Y^u=m-1)}{m+k}.
\end{aligned}
\tag{22}
$$

Here, we use the total expectation theorem and apply the same approximations as done in Eq.7.

### A.1.3 *Truncated Precision (TP)*.
The expected TP with respect to the interaction variable $Y$ is computed as follows:

$$
\begin{aligned}
\mathbb{E}_Y[U_{\text{TP}}(\pi_k|u)] &= \sum_{m=1}^{|\mathcal{I}_u^-|} P(S_Y^u = m)\, \mathbb{E}_{Y|S_Y^u=m}[U_{\text{TP}}(\pi_k|u)] \\
&= \sum_{m=1}^{|\mathcal{I}_u^-|} P(S_Y^u = m) \frac{\sum_{r=1}^{k} P(Y_{u,r}=1|S_Y^u=m)}{\min(k,m)} \\
&= \sum_{m=1}^{|\mathcal{I}_u^-|} \frac{\sum_{r=1}^{k} P(Y_{u,r}=1)P(S_{Y_{-r}}^u=m-1)}{\min(k,m)} \\
&\approx \sum_{m=1}^{M} \frac{\sum_{r=1}^{k} P(Y_{u,r}=1)P(S_Y^u=m-1)}{\min(k,m)}.
\end{aligned}
\tag{23}
$$

Here, we use the total expectation theorem and apply the same approximations as done in Eq.7.

## A.2 PerK on Multi-Domain Scenario

In the multi-domain scenario, PerK generates top-personalized-$K$ recommendations for each domain by slightly modifying Eq.3:

$$
\begin{aligned}
\{k_{\max}^x\}_{x\in\mathcal{S}} &= \operatorname*{argmax}_{\{k^x\}_{x\in\mathcal{S}}} \sum_{x\in\mathcal{S}} U(\pi_{k^x}|u,x) \\
\text{s.t.}\quad \pi_{k^x} &= \operatorname*{argsort}_{i\in\mathcal{I}_{u,x}^-} f_\theta^x(u,i)[:k],\quad \forall k^x \in [K], x\in\mathcal{S} \\
&\sum_{x\in\mathcal{S}} k^x \le N
\end{aligned}
\tag{24}
$$

$k^x$ is the recommendation size for domain $x$ and $\mathcal{S}$ is set of all domains. $U(\pi_{k^x}|u,x)$ is user utility of user $u$ and domain $x$. $\mathcal{I}_{u,x}^-$ is the unobserved itemset of user $u$ and domain $x$, $f_\theta^x$ is the recommender model for domain $x$ (it can be any cross/multi-domain recommender model). $N$ is the total recommendation size and $\mathbb{N}$ is the natural numbers set. To briefly explain, PerK finds the recommendation size for each domain to maximize the average user utility across all domains under the constraint that the total recommendation size should not exceed $N$. This optimization problem is a variant of the Knapsack problem [48] and can be solved by dynamic programming. We present a case study of the multi-domain scenario on the four largest domains of Amazon datasets in Section 6.6.

## A.3 Detailed Experiment Setup

We will publicly provide the GitHub repository of this work in the final version.

*A.3.1* **Dataset Statistics***.* Data statistics after the preprocessing are presented in Table 4.

**Table 4: Data statistics after the preprocessing.**

| Dataset | #Users | #Items | #Interactions | Sparsity |
|---------|--------|--------|---------------|----------|
| MovieLens 10M | 69,838 | 8,939 | 9,985,038 | 98.40% |
| CiteULike | 3,277 | 16,807 | 178,187 | 99.68% |
| MovieLens 25M | 162,414 | 18,424 | 24,811,113 | 99.17% |
| Amazon Books | 157,809 | 112,048 | 8,460,428 | 99.95% |

*A.3.2* **Training of Methods Compared***.* For AttnCut and MtCut, we use the source code of the authors.[6] We use a $K$-dimentional ranking score vector for the input of the models. We have tried to use the item embeddings as additional features for the models, but it did not increase the performance. For each dataset, hyperparameters are tuned by using grid searches on the validation set. We use Adam optimizer with learning rate in $\{10^{-2}, 10^{-3}, 10^{-4}\}$ and weight decay in $\{0, 10^{-1}, 10^{-3}, 10^{-5}, 10^{-7}, 10^{-9}\}$. We set the batch size to 64 and the embedding size to 64. The number of layers and the number of transformer heads are chosen from $\{1, 2\}$ and the dropout ratio is set to 0.2. Each model is trained until the convergence of validation performance.

*A.3.3* **Training of Base Recommender Models***.* We adopt three widely-used recommender models with various model architectures and loss functions. Bayesian Personalized Ranking (BPR) [44] captures the user-item relevance by the inner product of the user and the item embeddings, and is trained with the loss function that makes the model put the higher ranking score on the observed pair than the unobserved pair. Neural Collaborative Filtering (NCF) [14] adopts the feed-forward neural networks to output the ranking score of a user-item pair and is trained with the binary cross-entropy loss. LightGCN (LGCN) [13] adopts simplified Graph Convolutional Networks (GCN) [21] to capture the high-order interaction between the user and the item, and is trained with the loss function of BPR. Since PerK is a model-agnostic framework, other models can be also adopted for PerK in future work.

For all the base recommender models, we basically follow the source code of the authors and use PyTorch [39] for the implementation. For each dataset, hyperparameters are tuned by using grid searches on the validation set. We use Adam optimizer [20] with learning rate in $\{10^{-2}, 10^{-3}, 10^{-4}\}$ and weight decay in $\{0, 10^{-1}, 10^{-3}, 10^{-5}, 10^{-7}, 10^{-9}\}$. We set the batch size to 8192 and the embedding size is chosen from $\{64, 128\}$ for all base models. For NCF and LGCN, the number of layers is chosen from $\{1, 2\}$. The negative sample rate is set to 1 for all models. Each model is trained until the convergence of validation performance.

---

[6]https://github.com/Woody5962/Ranked-List-Truncation