# OpenReview forum: "Top-Personalized-K Recommendation"
_ACM.org/TheWebConf/2024/Conference — TheWebConf24_

### Official Review · Reviewer_uRD6 · 2023-11-20

**Novelty:** 6
**Technical Quality:** 6

**Review:**

# Summary

The paper presents a new technique to determine the ideal number of recommended items for each user. The authors argue that fixed list sizes are not optimal as they may either include irrelevant items or limit exposure to relevant ones. The problem is treated as an optimization problem over a general user utility function from which various evaluation metrics (such as NDCG and F1) can be derived. Since there is no ground truth for the unobserved user-item interactions, the objective function cannot be optimized directly. Instead, they use the Expected User Utility that can be estimated by treating the interaction labels for unobserved items as Bernoulli random variables. The Expected User Utility is computed over calibrated interaction probabilities. Experiments are conducted on several datasets, considering different baselines, where the proposed method shows superior results over the baselines.

# Clarity, Originality, and Significance

The paper is well-written, and the methodology is sound. The motivation is appealing, and the main ideas are built on established research. For example, the Expected User Utility comes from [45] and [46], while the calibration of Interaction Probability through Platt scaling comes from [12], [11], [34], [10], and [24]. The latter method was adapted to calibrate at the user level. The method and results have a high potential impact on academia and industry.

# Pros

- Paper is well written and technically sound;
- The method is built on sound principles and previous research;
- The experiments include several datasets and baselines;
- An ablation study to isolate the impact of calibration is included, showing that user-wise calibration indeed helps;
- The overall results are encouraging.

# Cons

- The recommendation task involves implicit feedback data, but some of the chosen datasets are rather explicit feedback transformed to accommodate the task. For example, two versions of MovieLens are used, while many pure implicit feedback data available could be better suited to this task. Random train/test splits might also be an issue. See "Balázs Hidasi and Ádám Tibor Czapp. 2023. Widespread Flaws in Offline Evaluation of Recommender Systems. In Proceedings of the 17th ACM Conference on Recommender Systems (RecSys '23)" for common problems in offline evaluation and how to mitigate them.
- I understand that the proposed method is agnostic to the recommendation model. Still, I would try to include more recommendation models, especially those appearing at well-known benchmarks, for this recommendation task. Please refer to "Steffen Rendle, Walid Krichene, Li Zhang, and Yehuda Koren. 2022. Revisiting the Performance of iALS on Item Recommendation Benchmarks. In Proceedings of the 16th ACM Conference on Recommender Systems (RecSys '22)."

**Questions:**

In Fig.2, for MovieLens 25M, there is a curious peak for k=50 in all cases. I wonder whether it is realistic to consider that the best choice is to recommend 50 items to a significant fraction of the users in any real-world application. Could the authors elaborate on that observation?

What would happen with non-personalized baselines? Would we observe the same gains?

**Reviewer Confidence:**

4: The reviewer is certain that the evaluation is correct and very familiar with the relevant literature

**Scope:**

4: The work is relevant to the Web and to the track, and is of broad interest to the community

---

### Official Review · Reviewer_fsvR · 2023-11-22

**Novelty:** 5
**Technical Quality:** 5

**Review:**

Summary:
- The paper discusses limitations of fixed-size top-K recommendation systems, proposing Top-Personalized-K Recommendation for personalized-sized lists. The PerK model, leveraging calibrated interaction probabilities, maximizes user satisfaction by selecting an optimal recommendation size. Extensive experiments show PerK's superiority, suggesting its potential for enhancing various real-world recommendation scenarios.

Pros:
1. This paper proposes a very interesting Top-Personalized-K recommendation task. The idea is simple and makes sense.
2. The paper is well-written and easy to follow.
3. The experimental results validate the efficacy of proposed PerK method.

**Questions:**

In real-world applications, if a screen can display 10 items, but the user's personalized k is 3, what should we do with the remaining space?

**Reviewer Confidence:**

2: The reviewer is willing to defend the evaluation, but it is likely that the reviewer did not understand parts of the paper

**Scope:**

4: The work is relevant to the Web and to the track, and is of broad interest to the community

---

### Official Review · Reviewer_k6gS · 2023-11-23

**Novelty:** 5
**Technical Quality:** 5

**Review:**

## summary:

This paper challenges the conventional fixed-size top-K recommendation
approach, arguing that it may not optimize user satisfaction. The
authors introduce the Top-Personalized-K Recommendation task, where
the recommendation size varies for each user to maximize individual
satisfaction. They propose the PerK framework, a model-agnostic
solution that estimates expected user utility using calibrated
interaction probabilities. PerK selects the recommendation size that
maximizes this expected utility. The framework involves a bi-level
optimization problem, using the concept of expected user utility and
calibrated interaction probabilities obtained through user-wise
calibration.

## strengths:
* s1. PerK is designed to seamlessly collaborate with any pre-trained
  recommender model, offering flexibility and adaptability to diverse
  recommendation scenarios. PerK is highly adaptable, suitable for
  integration into any item recommendation scenario with existing
  recommenders. Its simplicity doesn't compromise effectiveness.

* s2. The theoretical derivations are meticulously presented,
  contributing to a thorough understanding of the model.

* s3. The model undergoes a comprehensive evaluation with a detailed
  complexity analysis and AB testing conducted on an Amazon dataset.

* s4. Outperforming all baseline models by approximately 7% across four
  public datasets, the proposed model demonstrates robust performance.


## weaknesses:

* w1. The authors used three base recommender systems: Bayesian Personalized Ranking (BPR), Neural Collaborative Filtering (NCF), and LightGCN (LGCN). Notably, the authors did not provide explicit justification for the selection of these three base recommenders. Additionally, these particular models were not employed as base recommenders in baseline papers. The performance of this proposed model on top of advanced recommender systems remains uncertain, and the potential computational resources, particularly for large-scale recommendation systems, may necessitate significant resources for optimizing such a system.

**Questions:**

see weakness above.

**Reviewer Confidence:**

3: The reviewer is confident but not certain that the evaluation is correct

**Scope:**

4: The work is relevant to the Web and to the track, and is of broad interest to the community

---

### Official Review · Reviewer_F7hN · 2023-11-27

**Novelty:** 3
**Technical Quality:** 3

**Review:**

This paper proposed a Top-Personalized-K Recommendation, a new recommendation task resolving the limitation of the top-K recommendation with a globally fixed-size recommended list.

Pros:
1. The motivation is clearly clarified and the solution is well formulated.
2. Extensive experiments are conducted to demonstrate the performance and rationality of the proposed framework.

Cons:
1. In my opinion, this motivation is not reasonable in practice. The recommendation size K is constrained by business design rather than the algorithms. The example in Fig 1 seems unreasonable, the irrelevant items in the tail can not prove the impact of the personalized size, maybe just a result of the suboptimal ranking models.
2. The understanding of the application scenarios seems inappropriate. E.g. Multi-domain recommendation is a topic aiming at recommending items with a unified model which unitizes the mixed data from various domains. What you want to express here may be the page-level recommendation like[1].
3. The comparison in the experiments is not convincing enough for the unequal recommendation list.

[1] https://arxiv.org/pdf/2211.09303.pdf

**Questions:**

Please refer to the Cons.

**Ethics Review Description:**

There is no ethics issue

**Reviewer Confidence:**

3: The reviewer is confident but not certain that the evaluation is correct

**Scope:**

3: The work is somewhat relevant to the Web and to the track, and is of narrow interest to a sub-community

---

### Decision · Program_Chairs · 2024-01-22

**Decision:**

Accept

**Comment:**

This paper introduces Top-Personalized-K Recommendation, an innovative approach to recommendation tasks that addresses the constraints of traditional top-K recommendations, which rely on a universally fixed-size list. The reviewers find the idea of the paper intriguing and the experiments persuasive. They advise the authors to clearly justify their choice of the three base recommenders used, and to expand their research by conducting additional experiments with a broader range of recommendation models.